# Communication-efficient Distributed SGD with Sketching

**Nikita Ivkin** [*][†]
Amazon
ivkin@amazon.com

**Daniel Rothchild** [*]
UC Berkeley
drothchild@berkeley.edu

**Enayat Ullah** [*]
Johns Hopkins University
enayat@jhu.edu

**Vladimir Braverman** [‡]
Johns Hopkins University
vova@cs.jhu.edu

**Ion Stoica**
UC Berkeley
istoica@berkeley.edu

**Raman Arora**
Johns Hopkins University
arora@cs.jhu.edu

## Abstract

Large-scale distributed training of neural networks is often limited by network band-width, wherein the communication time overwhelms the local computation time. Motivated by the success of sketching methods in sub-linear/streaming algorithms, we introduce SKETCHED-SGD[4], an algorithm for carrying out distributed SGD by communicating sketches instead of full gradients. We show that SKETCHED-SGD has favorable convergence rates on several classes of functions. When considering all communication – both of gradients and of updated model weights – SKETCHED-SGD reduces the amount of communication required compared to other gradient compression methods from $\mathcal{O}(d)$ or $\mathcal{O}(W)$ to $\mathcal{O}(\log d)$, where $d$ is the number of model parameters and $W$ is the number of workers participating in training. We run experiments on a transformer model, an LSTM, and a residual network, demonstrating up to a 40x reduction in total communication cost with no loss in final model performance. We also show experimentally that SKETCHED-SGD scales to at least 256 workers without increasing communication cost or degrading model performance.

## 1 Introduction

Modern machine learning training workloads are commonly distributed across many machines using data-parallel synchronous stochastic gradient descent. At each iteration, $W$ worker nodes split a mini-batch of size $B$; each worker computes the gradient of the loss on its portion of the data, and then a parameter server sums each worker's gradient to yield the full mini-batch gradient. After using this gradient to update the model parameters, the parameter server must send back the updated weights to each worker. We emphasize that our method can naturally be extended to other topologies as well (e.g. ring, complete, etc.) – in particular we would then communicate sketches over a minimum spanning tree of the communication graph. However, for ease of exposition, in this work we focus exclusively on the star topology. For a fixed batch size $B$, the amount of data each worker processes – and therefore the amount of computation required – is inversely proportional to $W$. On the other hand, the amount of communication required per worker is independent of $W$. Even with optimal interleaving of the communication and computation, the total training time is at least the maximum

---

[*]equal contribution

[†]This work was done while the author was at Johns Hopkins University.

[‡]This work was done, in part, while the author was visiting the Simons Institute for the Theory of Computing.

[4]Code is available at https://github.com/dhroth/sketchedsgd

of the per-worker communication time and per-worker computation time. Increasing the number of workers $W$ therefore yields an increasingly marginal reduction in the training time, despite increasing the overall training cost (number of machines times training time) linearly in $W$.

Several approaches address this issue by using a large batch size to increase the per-worker computation time [You et al., 2017, Goyal et al., 2017]. However, theoretical and empirical evidence both suggest that there is a maximum mini-batch size beyond which the number of iterations required to converge stops decreasing, and generalization error begins to increase [Ma et al., 2017, Li et al., 2014, Golmant et al., 2018, Shallue et al., 2018, Keskar et al., 2016, Hoffer et al., 2017]. In this paper, we aim instead to decrease the communication cost per worker. We use a technique from streaming algorithms called sketching, which allows us to recover favorable convergence guarantees of vanilla SGD. In short, our algorithm has workers send gradient sketches of size $\mathcal{O}(\log d)$ instead of the gradients themselves. Although other methods for reducing the communication cost exist, to our knowledge ours is the only one that gives a per-worker communication cost that is sub-linear in $d$ and constant in $W$. In practice, we show that our method achieves high compression for large $d$ with no loss in model accuracy, and that it scales as expected to large $W$.

## 2 Related Work

Most existing methods for reducing communication cost in synchronous data-parallel distributed SGD either quantize or sparsify gradients. A number of quantization methods have been proposed. These methods either achieve only a constant reduction in the communication cost per iteration [Wen et al., 2017, Bernstein et al., 2018], or achieve an asymptotic reduction in communication cost per iteration at the expense of an equal (or greater) asymptotic increase in the number of iterations required [Alistarh et al., 2017]. Even in the latter case, the total communication required for all of training sees no asymptotic improvement.

Other methods sparsify the gradients instead of quantizing each gradient element [Stich et al., 2018, Alistarh et al., 2018, Lin et al., 2017]. A popular heuristic is to send the top-$k$ coordinates of the local worker gradients and then average them to obtain an approximate mini-batch gradient. These methods can achieve good performance in practice, but they suffer from a few drawbacks. They currently have no convergence guarantees, since the estimated mini-batch gradient can be very far from the true mini-batch gradient (unless explicitly assumed, as in e.g. Alistarh et al. [2018]), which precludes appealing to any known convergence result. Another drawback is that, although these methods achieve high compression rates when the workers transmit gradients to the parameter server, the return communication of the updated model parameters grows as $\mathcal{O}(W)$: the local top-$k$ of each worker may be disjoint, so there can be as many as $kW$ parameters updated each iteration. This $\mathcal{O}(W)$ communication cost is not just a technicality, since reducing the back-communication to $\mathcal{O}(k)$ would require sparsifying the sum of the local top-$k$, which could hinder convergence. Because of this scaling, local top-$k$ methods suffer from poor compression in settings with large $W$.

From another standpoint, all gradient compression techniques yield either biased or unbiased gradient estimates. A number of quantization methods are crafted specifically to yield unbiased estimates, such that the theoretical guarantees of SGD continue to apply [Alistarh et al., 2017, Wen et al., 2017]. However, even without these guarantees, a number of methods using biased gradient estimates were also found to work well in practice [Bernstein et al., 2018, Seide et al., 2014, Strom, 2015]. Recently, Stich et al. [2018], Karimireddy et al. [2019] gave convergence guarantees for this kind of biased compression algorithm, showing that accumulating compression error locally in the workers can overcome the bias in the weight updates as long as the compression algorithm obeys certain properties. Our method falls into this category, and we prove that compressing gradients with sketches obeys these properties and therefore enjoys the convergence guarantees in Stich et al. [2018]. In effect, we introduce a method that extends the theoretical results of Stich et al. [2018] from a single machine to the distributed setting. Concurrently with this work, Koloskova et al. [2019] also introduce a distributed learning algorithm with favorable convergence guarantees, in which workers communicate compressed gradients over an arbitrary network topology.

Prior work has proposed applying sketching to address the communication bottleneck in distributed and Federated Learning [Konečnỳ et al., 2016, Jiang et al., 2018]. However, these methods either do not have provable guarantees, or they apply sketches only to portions of the data, failing to alleviate the $\Omega(Wd)$ communication overhead. In particular, Konečnỳ et al. [2016] propose "sketched updates"

in Federated Learning for structured problems, and Jiang et al. [2018] introduce a range of hashing and quantization techniques to improve the constant in $\mathcal{O}(Wd)$.

Another line of work that we draw from applies sketching techniques to learning tasks where the model itself cannot fit in memory [Aghazadeh et al., 2018, Tai et al., 2018]. In our setting, we can afford to keep a dense version of the model in memory, and we only make use of the memory-saving properties of sketches to reduce communication between nodes participating in distributed learning.

## 3 Preliminaries

**SGD.** Let $w \in \mathbb{R}^d$ be the parameters of the model to be trained and $f_i(w)$ be the loss incurred by w at the $i^{\text{th}}$ data point $(x_i, y_i) \sim \mathcal{D}$. The objective is to minimize the generalization error $f(w) = \mathbb{E}_{(x_i, y_i) \sim \mathcal{D}}[f_i(w)]$. In large-scale machine learning, this objective is typically minimized using mini-batch stochastic gradient descent: given a step size $\eta_t$, at each iteration, w is updated as $w_{t+1} = w_t - \eta_t g_t$, where $g_t = \nabla_w \sum_{i \in \mathcal{M}} f_i(w)$ is the gradient of the loss computed on a minibatch $\mathcal{M}$. If $\mathcal{M}$ is randomly selected, then the gradient estimates $g_t$ are unbiased: i.e. $\mathbb{E}\left[g_t | \{w_i\}_{i=0}^{t-1}\right] = \nabla f(w_{t-1})$. As is standard, we further assume that the $g_t$ have bounded moment and variance: $\mathbb{E}\left[\|g_t\|_2^2 \,|\, \{w_i\}_{i=0}^{t-1}\right] \leq G^2$ and $\mathbb{E}\left[\|g_t - \nabla f(w_t)\|_2^2 \,|\, \{w_i\}_{i=0}^{t-1}\right] \leq \sigma^2$ for constants $G$ and $\sigma$. We adopt the usual definitions for smooth and strongly convex functions:

**Definition 1** (Smooth strongly convex function). *$f : \mathbb{R}^d \to \mathbb{R}$ is a L-smooth and $\mu$-strongly convex if the following hold $\forall w_1, w_2 \in \mathbb{R}^d$,*

1. *$\|\nabla f(w_2) - \nabla f(w_2)\| \leq L \|w_2 - w_1\|$ (Smoothness)*
2. *$f(w_2) \geq f(w_1) + \langle \nabla f(w_1), w_2 - w_1 \rangle + \frac{\mu}{2} \|w_2 - w_1\|^2$ (Strong convexity)*

For smooth strongly convex functions, SGD converges at a rate of $\mathcal{O}\left(\frac{G^2 L}{\mu T}\right)$ [Rakhlin et al., 2012].

**Count Sketch.** Our primary interest is in finding large coordinates (or "heavy hitters") of a gradient vector $g \in \mathbb{R}^d$. Heavy hitter sketches originated in the streaming model, where the vector g is defined by a sequence of updates $\{(i_j, w_j)\}_{j=1}^n$, such that the $j$-th update modifies the $i_j$-th coordinate of g as $g_{i_j}$ += $w_j$ [Charikar et al., 2002, Cormode and Muthukrishnan, 2005, Braverman et al., 2017]. In the streaming model, sketches must use memory sublinear in both $d$ and $n$.

In this work we compress a gradient vector g into a sketch $S(g)$ of size $O(\frac{1}{\varepsilon} \log d)$ using a Count Sketch [Charikar et al., 2002]. A Count Sketch $S(g)$ approximates every coordinate of g with an $\ell_2$ guarantee: it is always possible to recover $\hat{g}_i$ from $S(g)$ such that $g_i^2 - \varepsilon\|g\|_2^2 \leq \hat{g}_i^2 \leq g_i^2 + \varepsilon\|g\|_2^2$. In addition, $S(g)$ can approximate the $\ell_2$ norm of the entire gradient. These two properties let a sketch find every $\ell_2$ heavy hitter, i.e. every coordinate $i$ such that $g_i^2 > \varepsilon\|g\|_2^2$. With a small enough $\varepsilon$, the set of heavy hitters can be used as approximation of top-$k$ largest coordinates of gradient vector g.

Due to its linearity, the Count Sketch is widely adopted in distributed systems. Consider the case of a parameter server and two workers hosting vectors $g_1$ and $g_2$. To reduce communication, both workers can send the parameter server sketches $S(g_1)$ and $S(g_2)$ instead of the vectors themselves. The parameter server can then merge these sketches as $S(g) = S(g_1 + g_2) = S(g_1) + S(g_2)$. This lets the parameter server find the approximate top-$k$ largest coordinates in a vector distributed among many workers. We defer a more detailed discussion of the Count Sketch to Appendix C.

## 4 Sketched SGD

In SKETCHED-SGD, each worker transmits a sketch of its gradient instead of the gradient itself, as described above. The parameter server sums the workers' sketches, and then recovers the largest gradient elements by magnitude from the summed sketch. To improve the compression properties of sketching, we then perform a second round of communication, in which the parameter server requests the exact values of the top-$k$, and uses the sum of those in the weight update. This algorithm for recovering top-$k$ elements from a sketch is summarized in Algorithm 1.

Every iteration, only $k$ values of each worker's gradient are included in the final weight update. Instead of discarding the remaining $d - k$ gradient elements, it is important both theoretically and

empirically to accumulate these elements in local error accumulation vectors, which are then added to the next iteration's gradient [Karimireddy et al., 2019, Stich et al., 2018]. This process is summarized in Algorithm 2.

---

**Algorithm 1** HEAVYMIX

---

**Input:** S - sketch of gradient g; $k$ - parameter
1: Query $\hat{\ell}_2^2 = (1 \pm 0.5)\|g\|_2^2$ from sketch $S$
2: $\forall i$ query $\hat{g}_i^2 = g_i^2 \pm \frac{1}{2k}\|g\|_2^2$ from sketch $S$
3: $H \leftarrow \left\{ i | \hat{g}_i \geq \hat{\ell}_2^2/k \right\}$ and $NH \leftarrow \left\{ i | \hat{g}_i < \hat{\ell}_2^2/k \right\}$
4: $\text{Top}_k = H \cup \text{rand}_l(NH)$, where $l = k - |H|$
5: second round of communication to get exact values of $\text{Top}_k$
**Output:** $\tilde{g}$: $\forall i \in \text{Top}_k : \tilde{g}_i = g_i$ and $\forall i \notin \text{Top}_k : \tilde{g}_i = 0$

---

**Algorithm 2** SKETCHED-SGD

---

**Input:** $k, \xi, T, W$
1: $\eta_t \leftarrow \frac{1}{t+\xi}, q_t \leftarrow (\xi+t)^2, Q_T = \sum_{t=1}^T q_t, a_0 = 0$
2: **for** $t = 1, 2, \cdots T$ **do**
3:      Compute stochastic gradient $g_t^i$                                              Worker$_i$
4:      Error correction: $\bar{g}_t^i = \eta_t g_t^i + a_{t-1}^i$                             Worker$_i$
5:      Compute sketches $S_t^i$ of $\bar{g}_t^i$ and send to Parameter Server        Worker$_i$
6:      Aggregate sketches $S_t = \frac{1}{W}\sum_{i=1}^W S_t^i$                    Parameter Server
7:      $\tilde{g}_t = \text{HEAVYMIX}(S_t, k)$                              Parameter Server
8:      Update $w_{t+1} = w_t - \tilde{g}_t$ and send $\tilde{g}_t$ (which is $k$-sparse) to Workers    Parameter Server
9:      Error accumulation: $a_t^i = \bar{g}_t^i - \tilde{g}_t$                         Worker$_i$
10: **end for**
**Output:** $\hat{w}_T = \frac{1}{Q_T}\sum_{t=1}^T q_t w_t$

---

We now state convergence results for SKETCHED-SGD. Proofs are deferred to Appendix A.

**Theorem 1** (strongly convex, smooth). *Let $f : \mathbb{R}^d \to \mathbb{R}$ be a L-smooth $\mu$-strongly convex function, and let the data be shared among $W$ workers. Given $0 < k \leq d, 0 < \alpha, and \delta < 1$, Algorithm 2* SKETCHED-SGD *run with sketch size $= \mathcal{O}\left(k \log(dT/\delta)\right)$, step size $\eta_t = \frac{1}{t+\xi}$, with $\xi > 2 + \frac{d(1+\beta)}{k(1+\rho)}$, with $\beta > 4$ and $\rho = \frac{4\beta}{(\beta-4)(\beta+1)^2}$ after $T$ steps outputs $\hat{w}_T$ such that the following holds,*

1. *With probability at least $1 - \delta$, $\mathbb{E}\left[f(\hat{w}_T)\right] - f(w^*) \leq \mathcal{O}\left(\frac{\sigma^2}{\mu T} + \frac{d^2 G^2 L}{k^2 \mu^2 T^2} + \frac{d^3 G^3}{k^3 \mu T^3}\right)$*

2. *The total communication per update is $\Theta(k \log(dT/\delta)W)$ bits.*

**Remarks**

1. The convergence rate for vanilla SGD is $\mathcal{O}(1/T)$. Therefore, our error is larger the SGD error when $T = o((d/k)^2)$, and approaches the SGD error for $T = \Omega((d/k)^2)$.
2. Although not stated in this theorem, Stich et al. [2018] show that using the top-$k$ coordinates of the true mini-batch gradient as the SGD update step yields a convergence rate equivalent to that of SKETCHED-SGD. We therefore use this "true top-$k$" method as a baseline for our results.
3. Note that the leading term in the error is $O(\sigma^2/T)$ (as opposed to $O(G^2/T)$ in [Stich et al., 2018]); this implies that in setting where the largest minibatch size allowed is too large to fit in one machine, and going distributed allows us to use larger mini-batches, the variance reduces by a factor $W$. This reduces the number of iterations required (asymptotically) linearly with $W$.
4. As is standard, the above high probability bound can be converted to an expectation (over randomness in sketching) bound; this is stated as Theorem 6 in the Appendix A.
5. The result of [Karimireddy et al., 2019] allows us to extend our theorems to smooth nonconvex and non-smooth convex functions; these are presented as Theorems 4 and 5 in the Appendix B..

**Proof Sketch.** The proof consists of two parts. First, we show that SKETCHED-SGD satisfies the criteria in Stich et al. [2018], from which we obtain a convergence result when running SKETCHED-SGD on a single machine. We then use properties of the Count Sketch to extend this result to the distributed setting.

For the first part, the key idea is to show that our heavy hitter recovery routine HEAVYMIX satisfies a *contraction* property, defined below.

**Definition 2** ($\tau$-contraction [Stich et al., 2018]). *A $\tau$-contraction operator is a possibly randomized operator* $comp : \mathbb{R}^d \to \mathbb{R}^d$ *that satisfies:* $\forall \mathrm{x} \in \mathbb{R}^d$, $\mathbb{E}\left[\|\mathrm{x} - comp(\mathrm{x})\|^2\right] \leq (1-\tau)\|\mathrm{x}\|^2$

Given a contraction operator with $\tau = k/d$, and assuming that the stochastic gradients g are unbiased and bounded as $\mathbb{E}\left[\|\mathrm{g}\|^2\right] \leq G^2$, choosing the step-size appropriately, Stich et al. [2018] give a convergence rate of $\mathcal{O}\left(\frac{G^2}{\mu T} + \frac{d^2 G^2 L}{k^2 \mu^2 T^2} + \frac{d^3 G^3}{k^3 \mu T^3}\right)$ for sparsified SGD with error accumulation. As stated in Lemma 1, HEAVYMIX satisfies this contraction property, and therefore inherits this (single-machine) convergence result:

**Lemma 1.** HEAVYMIX, *with sketch size* $\Theta(k \log(d/\delta))$ *is a* $k/d$-*contraction with probability* $\geq 1-\delta$.

This completes the first part of the proof. To extend SKETCHED-SGD to the distributed setting, we exploit the fact that Count Sketches are linear, and can approximate $\ell_2$ norms. The full proof is deferred to Appendix A.

# 5 Empirical Results

## 5.1 Training Algorithm

In practice, we modify SKETCHED-SGD in the following ways

- We employ momentum when training. Following Lin et al. [2017], we use momentum correction and momentum factor masking. Momentum factor masking mitigates the effects of stale momentum, and momentum correction is a way to do error feedback on SGD with momentum [Karimireddy et al., 2019].
- We use the Count Sketch to identify heavy coordinates, however we perform an additional round of communication to collect the exact values of those coordinates. In addition, to identify the top $k$ heavy coordinates, we query the Count Sketch, and then each of the workers, for the top $Pk$ elements instead; this is a common technique used with sketching to improve stability. The total resulting communication cost is $Pk + |S| + k$ per worker, where $|S|$ is the size of the sketch, and the last $k$ corresponds to the the updated model parameters the parameter server must send back to the workers.
- We transmit gradients of the bias terms uncompressed. The number of bias terms in our models is $< 1\%$ of the total number of parameters.

Our emperical training procedure is summarized in Algorithm 3.

---

**Algorithm 3** EMPIRICAL TRAINING

**Input:** $k, \eta_t, m, T$
1: $\forall i : \mathrm{u}^i, \mathrm{v}^i \leftarrow 0$
2: Initialize $w_0^i$ from the same random seed on each Worker.
3: **for** $t = 1, 2, \ldots T$ **do**
4:      Compute stochastic gradient $\mathrm{g}_t^i$                Worker$_i$
5:      Momentum: $\mathrm{u}^i \leftarrow m\mathrm{u}^i + \mathrm{g}_t^i$            Worker$_i$
6:      Error accumulation: $\mathrm{v}^i \leftarrow \mathrm{v}^i + \mathrm{u}^i$        Worker$_i$
7:      Compute sketch $\mathrm{S}_t^i$ of $\mathrm{v}^i$ and send to Parameter Server           Worker$_i$
8:      Aggregate sketches $\mathrm{S}_t = \frac{1}{W}\sum_{i=1}^{W} \mathrm{S}_t^i$        Parameter Server
9:      Recover the top-$Pk$ coordinates from $\mathrm{S}_t$: $\tilde{\mathrm{g}}_t = top_{Pk}(S_t)$      Parameter Server
10:     Query all workers for exact values of nonzero elements in $\tilde{\mathrm{g}}_t$; store the sum in $\tilde{\mathrm{g}}_t$      Parameter Server
11:     Send the $k$-sparse $\tilde{\mathrm{g}}_t$ to Workers        Parameter Server
12:     update $\mathrm{w}_{t+1}^i = \mathrm{w}_t^i - \eta_t \tilde{\mathrm{g}}_t$ on each worker        Worker$_i$
13:     $\mathrm{u}^i, \mathrm{v}^i \leftarrow 0$, for all $i$ s.t. $\tilde{\mathrm{g}}_t^i \neq 0$        Worker$_i$
14: **end for**

---

## 5.2 Sketching Implementation

We implement a parallelized Count Sketch with PyTorch [Paszke et al., 2017]. The Count Sketch data structure supports a query method, which returns a provable $\pm\varepsilon\|\mathrm{g}\|_2$ approximation to each

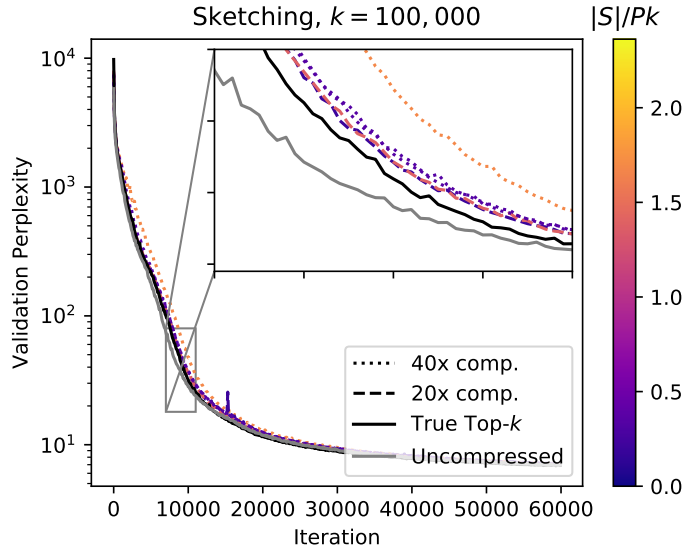

Figure 1: Learning curves for a transformer model trained on the WMT 2014 English to German translation task. All models included here achieve comparable BLEU scores after 60,000 iterations (see Table 1). Each run used 4 workers.

coordinate value. However, to the best of our knowledge, there is no efficient way to find heavy coordinates in the presence of negative inputs. Fortunately, in our application, it is computationally efficient on the GPU to simply query the sketch for every gradient coordinate, and then choose the largest elements.

### 5.3 Large $d$

First, we show that SKETCHED-SGD achieves high compression with no loss in accuracy. Because the sketch size grows as $\mathcal{O}(\log d)$, we expect to see the greatest compression rates for large $d$. Accordingly, we test on a transformer model with 90M parameters, and on a stacked LSTM model with 73M parameters. We train both models on the WMT 2014 English to German translation task, and we use code from the OpenNMT project [Klein et al., 2017]. In all cases, the compression factor for SKETCHED-SGD is computed as $2d/(|S| + Pk + k)$, where $2d$ is the cost to send a (dense) gradient and receive a new (dense) parameter vector, $|S|$ is the sketch size, $Pk$ is the number of elements sent in the second round of communication, and the last $k$ represents the number of modified parameter values that must be sent back to each worker.

SKETCHED-SGD achieves the same theoretical convergence rate as top-$k$ SGD, in which the weight update consists of the top-$k$ elements of the full mini-batch gradient. We therefore perform experiments with SKETCHED-SGD using a value of $k$ that yields good performance for top-$k$ SGD. Figure 2 shows top-$k$ results over a range of values of $k$. Curiously, performance starts to degrade for large $k$. Although performance on the training data should in principle strictly improve for larger $k$, sparsifying gradients regularizes the model, so $k < d$ may yield optimal performance on the test set. In addition, we expect performance to degrade on both the training and test sets for large $k$ due to momentum factor masking. To mitigate stale momentum updates, momentum factor masking zeros the velocity vector at the $k$ coordinates that were updated in each iteration. In the limit $k = d$, this completely negates the momentum, hindering convergence. For all SKETCHED-SGD experiments on these two models, we use $k = 100,000$, for which top-$k$ SGD yields a BLEU score of 26.65 for the transformer and 22.2 for the LSTM. For reference, uncompressed distributed SGD with the same hyperparameters achieves a BLEU of 26.29 for the transformer and 20.87 for the LSTM. Using SKETCHED-SGD, we can obtain, with no loss in BLEU, a 40x reduction in the total communication cost during training, including the cost to disseminate updated model parameters. See Table 1 for a summary of BLEU results. Compression numbers include both the communication required to send gradients as well as the cost to send back the new model parameters. We do not include the cost to

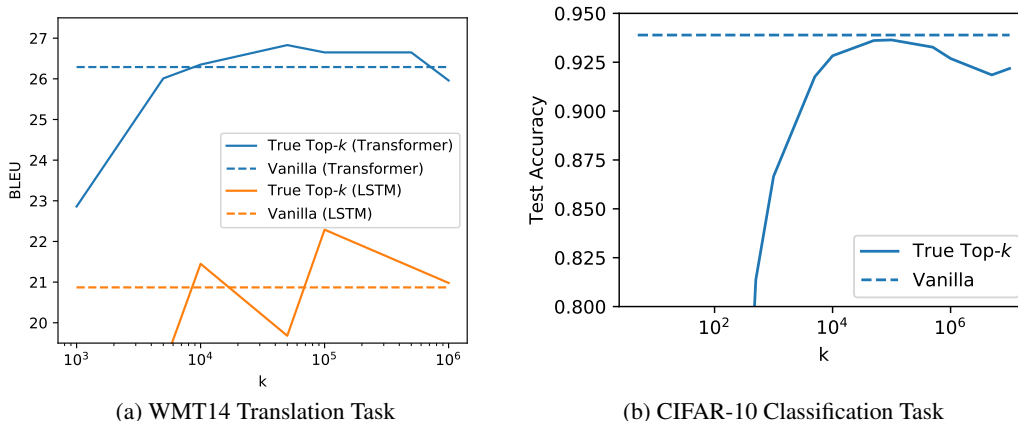

| | (a) WMT14 Translation Task | (b) CIFAR-10 Classification Task |
|---|---|---|

Figure 2: True top-$k$ results for a range of $k$. Left: two models (transformer and LSTM) on the WMT 2014 English to German translation task. Right: a residual network on the CIFAR-10 classification task. For the larger models (left), true top-$k$ slightly outperforms the baseline for a range of $k$. We suspect this is because $k$-sparsifying gradients serves to regularize the model.

| | BLEU (transformer) | BLEU (LSTM) |
|---|---|---|
| Uncompressed Distributed SGD | 26.29 | 20.87 |
| Top-$100,000$ SGD | 26.65 | 22.2 |
| SKETCHED-SGD, 20x compression | 26.87[5] | – |
| SKETCHED-SGD, 40x compression | 26.79[6] | 20.95 [7] |

Table 1: BLEU scores on the test data achieved for uncompressed distributed SGD, top-$k$ SGD, and SKETCHED-SGD with 20x and 40x compression. Compression rates represent the total reduction in communication, including the cost to transmit the updated model parameters. Larger BLEU score is better. For both models, top-$k$ SGD with $k = 100,000$ achieves a higher BLEU score than uncompressed distributed SGD. This difference may be within the error bars, but if not, it may be that stepping in only the direction of the top-$k$ is serving as a regularizer on the optimizer. Our main experiments are on the transformer model, for which we run additional experiments using 20x compression that we did not complete for the LSTM model.

request the $Pk$ coordinates, nor to specify which $k$ model parameters have been updated, since these quantities can be efficiently coded, and contribute little to the overall communication.

Given that our algorithm involves a second round of communication in which $Pk$ gradient elements are transmitted, we investigate the tradeoff between a large sketch size and a large value of $P$. Approaching a sketch size of zero corresponds to using a weight update that is the top-$k$ of a randomly chosen set of $Pk$ gradient coordinates. Experiments with extremely small sketch size $|S|$ or extremely small values of $P$ tended to diverge or achieve very low BLEU score. For values of $|S|/Pk$ closer to 1, we plot learning curves in Figure 1. As expected, uncompressed SGD trains fastest, followed by top-$k$ SGD, then 20x compression SKETCHED-SGD, then 40x compression SKETCHED-SGD. For the two 20x compression runs, the ratio of the sketch size to the number of exact gradient values computed has little effect on convergence speed. However, the higher compression runs prefer a relatively larger value of $P$.

### 5.4 Large $W$

To re-iterate, the per-worker communication cost for SKETCHED-SGD is not only sub-linear in $d$, but also independent of $W$. To demonstrate the power of this experimentally, we train a residual

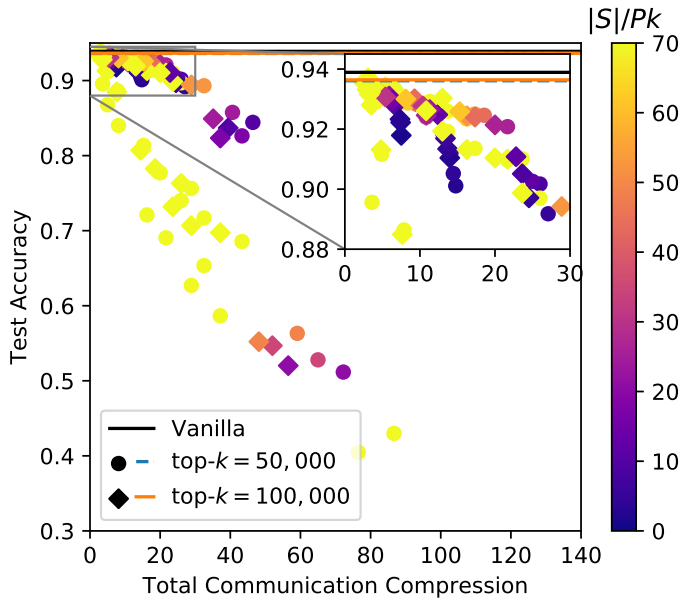

Figure 3: Tradeoff between compression and model accuracy for a residual network trained on CIFAR-10. We show results for $k = 50,000$ as well as $k = 100,000$, and color code each trained model based on the ratio of sketch size to the cost of the second round of communication. The (nearly overlapping) solid orange and dashed blue lines show the accuracy achieved by top$-k$ SGD for the two values of $k$, and the black line shows the accuracy achieved by uncompressed distributed SGD. All models in this plot were trained with 4 workers.

network on the CIFAR-10 dataset with SKETCHED-SGD, using up to 256 workers [Krizhevsky and Hinton, 2009]. We compare to local top-$k$, a method where each worker computes and transmits only the top-$k$ elements of its gradient. The version of local top-$k$ SGD we compare to is similar to Deep Gradient Compression, except we do not clip gradients, and we warm up the learning rate instead of the sparsity [Lin et al., 2017]. Results are shown in Figure 4. Neither algorithm sees an appreciable drop in accuracy with more workers, up to $W = 256$. However, while the communication cost of SKETCHED-SGD is constant in $W$, the communication cost for local top-$k$ scales with $W$ until reaching $\Theta(d)$. This scaling occurs because the local top-$k$ of each worker might be disjoint, leading to as many as $kW$ parameters being updated. In practice, we do in fact observe nearly linear scaling of the number of parameters updated each iteration, until saturating at $d$ (dashed orange line in Figure 4). For $W = 256$, the communication of the updated model parameters back to each worker is nearly dense ($d \approx 6.5 \times 10^6$), reducing the overall compression of local top-$k$ to at best $\sim 2\times$.

For a fixed small number of workers ($W = 4$), we also investigate the tradeoff between compression rate and final test accuracy. Figure 3 shows this tradeoff for two values of $k$ and a wide range of sketch sizes and values of $P$. As expected, increasing the compression rate leads to decreasing test accuracy. In addition, as evidenced by the color coding, using a very large sketch size compared to $Pk$ tends to yield poor results. Although high compression rates decrease accuracy, in our experience, it is possible to make up for this accuracy drop by training longer. For example, choosing one of the points in Figure 3, training with 17x compression for the usual number of iterations gives 92.5% test accuracy. Training with 50% more iterations (reducing to 11x overall compression) restores accuracy to 94%. In Figure 3, every model is trained for the same number of iterations.

## 6 Discussion

In this work we introduce SKETCHED-SGD, an algorithm for reducing the communication cost in distributed SGD using sketching. We provide theoretical and experimental evidence that our method can help alleviate the difficulties of scaling SGD to many workers. While uncompressed distributed SGD requires communication of size $2d$, and other gradient compressions improve this to $\mathcal{O}(d)$

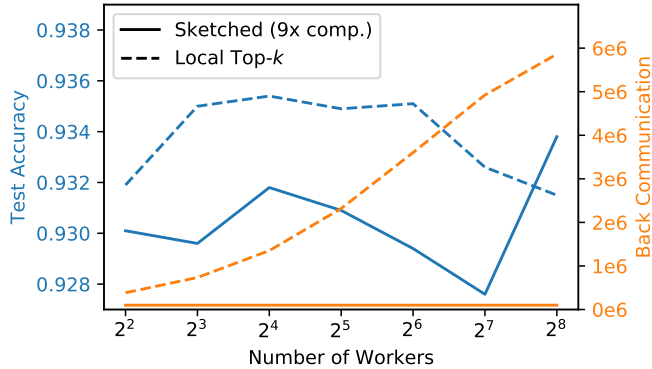

Figure 4: Comparison between SKETCHED-SGD and local top-$k$ SGD on CIFAR10. Neither algorithm sees an appreciable drop in performance for up to 256 workers, but the amount of communication required for local top-$k$ grows quickly to $\approx d = 6.5 \times 10^6$ as the number of workers increases. As a result, the best overall compression that local top-$k$ can achieve for many workers is 2x.

or $\mathcal{O}(W)$, SKETCHED-SGD further reduces the necessary communication to $\mathcal{O}(\log d)$. Besides reducing communication, our method provably converges at the same rate as SGD, and in practice we are able to reduce the total communication needed by up to 40x without experiencing a loss in model quality.

A number of other techniques for efficient training could be combined with SKETCHED-SGD, including gradient quantization and asynchronous updates. We expect that the advantages asynchronous updates bring to regular SGD will carry over to SKETCHED-SGD. And given that elements of gradient sketches are sums of gradient elements, we expect that quantizing sketches will lead to similar tradeoffs as quantizing the gradients themselves. Preliminary experiments show that quantizing sketches to 16 bits when training our ResNets on CIFAR-10 leads to no drop in accuracy, but we leave a full evaluation of combining quantization, as well as asynchronous updates, with SKETCHED-SGD to future work.

Machine learning models are constantly growing in size (e.g. OpenAI's GPT-2, a transformer with 1.5 billion parameters [Radford et al., 2019]), and training is being carried out on a larger and larger number of compute nodes. As communication increasingly becomes a bottleneck for large-scale training, we argue that a method that requires only $\mathcal{O}(\log d)$ communication has the potential to enable a wide range of machine learning workloads that are currently infeasible, from highly parallel training in the cloud, to Federated Learning at the edge [McMahan et al., 2016].

# 7 Acknowledgements

This research was supported, in part, by NSF BIGDATA grants IIS-1546482 and IIS-1838139, NSF CAREER grant 1652257, ONR Award N00014-18-1-2364 and the Lifelong Learning Machines program from DARPA/MTO. This material is based upon work supported by the National Science Foundation Graduate Research Fellowship under Grant No. DGE 1752814.

## Footnotes

[5]Sketch size: 5 rows by 1M columns; $P = 36$.

[6]Sketch size: 15 rows by 180,000 columns; $P = 16$.

[7]Sketch size: 5 rows by 180,000 columns, $P = 26$

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
