[Supplementary Material]

# Supplementary

## A  Proofs

*Proof of Lemma 1.* Given $g \in \mathbb{R}$, the HEAVYMIX algorithm extracts all $(1/k, \ell_2^2)$-heavy elements from a Count Sketch $S$ of $g$. Let $\hat{g}$ be the values of all elements recovered from its sketch. For a fixed $k$, we create two sets $H$ (heavy), and $NH$ (not-heavy). All coordinates of $\hat{g}$ with values at least $\frac{1}{k}\hat{\ell}_2^2$ are put in $H$, and all others in $NH$, where $\hat{\ell}_2$ is the estimate of $\|g\|_2$ from the Count Sketch. Note that the number of elements in $H$ can be at most $k$. Then, we sample uniformly at random $l = k - |H|$ elements from $NH$, and finally output its union with $H$. We then do a second round of communication to get exact values of these $k$ elements.

Note that, because of the second round of communication in HEAVYMIX and the properties of the Count Sketch, with probability at least $1 - \delta$ we get the exact values of all elements in $H$. Call this the "heavy hitters recovery" event. Let $g_H$ be a vector equal to $g$ at the coordinates in $H$, and zero otherwise. Define $g_{NH}$ analogously. Conditioning on the heavy hitters recovery event, and taking expectation over the random sampling, we have

$$\mathbb{E}\left[\|g - \tilde{g}\|^2\right] = \|g_H - \bar{g}_H\|^2 + \mathbb{E}\left[\|g_{NH} - \operatorname{rand}_l(g_{NH})\|^2\right]$$

$$\leq \left(1 - \frac{k - |H|}{d - |H|}\right)\|g_{NH}\|^2 \leq \left(1 - \frac{k - |H|}{d - |H|}\right)\left(1 - \frac{|H|}{2k}\right)\|g\|^2$$

Note that, because we condition on the heavy hitter recovery event, $\bar{g}_H = g_H$ due to the second round communication (line 9 of Algorithm 3). The first inequality follows using Lemma 1 from Stich et al. [2018]. The second inequality follows from the fact that the heavy elements have values at least $\frac{1}{k}\hat{\ell}_2^2 \geq \frac{1}{2k}\|g\|^2$, and therefore $\|g_{NH}\|^2 = \|g\|^2 - \|g_H\|^2 \leq \left(1 - \frac{|H|}{2k}\right)\|g\|^2$.

Simplifying the expression, we get

$$\mathbb{E}\left[\|g - \tilde{g}\|^2\right] \leq \left(\frac{2k - |H|}{2k}\right)\left(\frac{d - k}{d - |H|}\right)\|g\|^2 = \left(\frac{2k - |H|}{2k}\right)\left(\frac{d}{d - |H|}\right)\left(1 - \frac{k}{d}\right)\|g\|^2.$$

Note that the first two terms can be bounded as follows:

$$\left(\frac{2k - |H|}{2k}\right)\left(\frac{d}{d - |H|}\right) \leq 1 \iff kd - |H|d \leq kd - 2k|H| \iff |H|(d - 2k) \geq 0$$

which holds when $k \leq d/2$ thereby completing the proof.

$\square$

### A.1  Proof of the main theorem

*Proof of Theorem 1.* First note that, from linearity of sketches), the top-$k$ (or heavy) elements from the merged sketch $S_t = \sum_{i=1}^W S_t^i$ are the top-$k$ of the sum of vectors that were sketched. We have already shown in Lemma 1 that that extracting the top-$k$ elements from $S - T$ using HEAVYMIX gives us a $k$-contraction on the sum of gradients. Moreover since the guarantee is relative and norms are positive homogeneous, the same holds for the average, i.e. when dividing by $W$. Now since the average of stochastic gradients is still an unbiased estimate, this reduces to SKETCHED-SGD on one machine, and the convergence therefore follows from Theorem 2. $\square$

A key ingredient is the result in the one machine setting, stated below.

**Theorem 2.** *Let $f : \mathbb{R}^d \to \mathbb{R}$ be a $L$-smooth $\mu$-strongly convex function. Given $T > 0$ and $0 < k \leq d, 0 < \delta < 1$, and a $\tau_k$-contraction, Algorithm 2 SKETCHED-SGD with sketch size $\mathcal{O}\left(k\log(dT/\delta)\right)$ and step size $\eta_t = \frac{1}{t + \xi}$, with $\xi > 1 + \frac{1 + \beta}{\tau_k(1 + \rho)}$, with $\beta > 4$ and $\rho = \frac{4\beta}{(\beta - 4)(\beta + 1)^2}$, after $T$ steps outputs $\hat{w}_T$ such that with probability at least $1 - \delta$*

$$\mathbb{E}\left[f(\hat{w}_T)\right] - f(w^*) \leq \mathcal{O}\left(\frac{\sigma^2}{\mu T} + \frac{G^2 L}{\tau_k^2 \mu^2 T^2} + \frac{G^3}{\tau_k^3 \mu T^3}\right)$$

*Proof of Theorem 2.* The proof, as in Stich et al. [2018], just follows using convexity and Lemmas 3,2 and fact 3. The lemmas which are exactly same as Stich et al. [2018], are stated as facts. However, the proofs of lemmas, which change are stated in full for completeness, with the changes highlighted.

From convexity we have that

$$f\left(\frac{1}{Q_T}\sum_{i=1}^T q_t \mathrm{w}_t\right) - f(\mathrm{w}^*) \leq \frac{1}{Q_T}\sum_{t=1}^T q_t f(\mathrm{w}_t) - f(\mathrm{w}^*) = \frac{1}{Q_T}\sum_{t=1}^T q_t \left(f(\mathrm{w}_t) - f(\mathrm{w}^*)\right)$$

Define $\epsilon_t = f(\mathrm{w}_t) - f(\mathrm{w}^*)$, the excess error of iterate $t$. From Lemma 2 we have,

$$\mathbb{E}\left[\|\tilde{\mathrm{w}}_{t+1} - \mathrm{w}^*\|^2\right] \leq \left(1 - \frac{\eta_t \mu}{2}\right)\mathbb{E}\left[\|\tilde{\mathrm{w}}_t - \mathrm{w}^*\|^2\right] + \sigma^2\eta_t^2 - \left(1 - \frac{2}{\xi}\right)\epsilon_t\eta_t + (\mu + 2L)\mathbb{E}\left[\|\mathrm{a}_t\|^2\right]\eta_t$$

Bounding the last term using Lemma 3, with probability at least $1 - \delta$, we get,

$$\mathbb{E}\left[\|\tilde{\mathrm{w}}_{t+1} - \mathrm{w}^*\|^2\right] \leq \left(1 - \frac{\eta_t \mu}{2}\right)\mathbb{E}\left[\|\tilde{\mathrm{w}}_t - \mathrm{w}^*\|^2\right] + \sigma^2\eta_t^2 - \left(1 - \frac{2}{\xi}\right)\epsilon_t\eta_t + \frac{(\mu + 2L)4\beta G^2}{\tau_k^2(\beta - 4)}\eta_t^3$$

where $\tau_k$ is the contraction we get from HEAVYMIX. We have alreay show that $\tau_k \leq \frac{k}{d}$.

Now using Lemma 3 and the fist equation, we get,

$$f\left(\frac{1}{Q_T}\sum_{i=1}^T q_t \mathrm{w}_t\right) - f(\mathrm{w}^*) \leq \frac{\mu\xi^4\mathbb{E}\left[\|\mathrm{w}_0 - \mathrm{w}^*\|^2\right]}{8(\xi - 2)Q_T} + \frac{4T(T + 2\xi)\xi\sigma^2}{\mu(\xi - 2)Q_T} + \frac{256(\mu + 2L)\beta\xi G^2 T}{\mu^2(\beta - 4)\tau_k^2(\xi - 2)Q_T}$$

Note that $\xi > 2 + \frac{1+\beta}{\tau_k(1+\rho)}$. Moreover $Q_T = \sum_{t=1}^T q_t = \sum_{t=1}^T (\xi + t)^2 \geq \frac{1}{3}T^3$ upon expanding and using the conditions on $\xi$. Also $\xi/(\xi - 2) = \mathcal{O}\left(1 + 1/\tau_k\right)$.

Finally using $\sigma^2 \leq G^2$ and Fact 1 to bound $\mathbb{E}\left[\|\mathrm{w}_0 - \mathrm{w}^*\|^2\right] \leq 4G^2/\mu^2$ completes the proof.

$\square$

**Lemma 2.** *Let $f : \mathbb{R}^d \to \mathbb{R}$ be a L-smooth $\mu$-strongly convex function, and $\mathrm{w}^*$ be its minima. Let $\{\mathrm{w}_t\}_t$ be a sequence of iterates generated by Algorithm 2.*

*Define error $\epsilon_t := \mathbb{E}\left[f(\mathrm{w}_t) - f(\mathrm{w}^*)\right]$ and $\tilde{\mathrm{w}}_{t+1} = \tilde{\mathrm{w}}_t - \eta_t g_t$ be a stochastic gradient update step at time t, with $\mathbb{E}\left[\|g_t - \nabla f(\mathrm{w}_t)\|^2\right] \leq \sigma^2$, $\mathbb{E}\left[\|g_t\|^2\right] \leq G^2$ and $\eta_t = \frac{1}{\mu(t+\xi)}, \xi > 2$ then we have,*

$$\mathbb{E}\left[\|\tilde{\mathrm{w}}_{t+1} - \mathrm{w}^*\|^2\right] \leq \left(1 - \frac{\eta_t \mu}{2}\right)\mathbb{E}\left[\|\tilde{\mathrm{w}}_t - \mathrm{w}^*\|^2\right] + \sigma^2\eta_t - \left(1 - \frac{2}{\xi}\right)\epsilon_t\eta_t + (\mu + 2L)\mathbb{E}\left[\|\mathrm{a}_t\|^2\right]\eta_t$$

*Proof.* This is the first step of the perturbed iterate analysis framework Mania et al. [2015]. We follow the steps as in Stich et al. [2018]. The only change is that the proof of Stich et al. [2018] works with bounded gradients i.e. $\mathbb{E}\left[\|g\|^2\right] \leq G^2$. This assumption alone, doesn't provide the variance reduction effect in the distributed setting. We therefore adapt the analysis with the the variance bound $\mathbb{E}\left[\|g - \nabla f(\mathrm{w})\|^2\right] \leq \sigma^2$.

$$\|\tilde{\mathrm{w}}_{t+1} - \mathrm{w}^*\|^2 = \|\tilde{\mathrm{w}}_{t+1} - \tilde{\mathrm{w}}_t + \tilde{\mathrm{w}}_t - \mathrm{w}^*\|^2 = \|\tilde{\mathrm{w}}_{t+1} - \tilde{\mathrm{w}}_t\|^2 + \|\tilde{\mathrm{w}}_t - \mathrm{w}^*\|^2 + 2\langle\tilde{\mathrm{w}}_t - \mathrm{w}^*, \tilde{\mathrm{w}}_{t+1} - \tilde{\mathrm{w}}_t\rangle$$
$$= \eta_t^2\|g_t\|^2 + \|\tilde{\mathrm{w}}_t - \mathrm{w}^*\|^2 + 2\langle\tilde{\mathrm{w}}_t - \mathrm{w}^*, \tilde{\mathrm{w}}_{t+1} - \tilde{\mathrm{w}}_t\rangle = \eta_t^2\|g_t - \nabla f(\mathrm{w}_t)\|^2 + \eta_t^2\|\nabla f(\mathrm{w}_t)\|^2 + \|\tilde{\mathrm{w}}_t - \mathrm{w}^*\|^2$$
$$+ 2\eta_t\langle g_t - \nabla f(\mathrm{w}_t), \nabla f(\mathrm{w}_t)\rangle + 2\eta_t\langle\mathrm{w}^* - \tilde{\mathrm{w}}_t, g_t\rangle$$

Taking expectation with respect to the randomness of the last stochastic gradient, we have that the term $\langle g_t - \nabla f(\mathrm{w}_t), \nabla f(\mathrm{w}_t)\rangle = 0$ by $\mathbb{E}[g_t] = \nabla f(\mathrm{w}_t)$. Moreover, the term $\mathbb{E}[g_t - \nabla f(\mathrm{w}_t)]^2 \leq \sigma^2$. We expand the last term as,

$$\langle\mathrm{w}^* - \tilde{\mathrm{w}}_t, \nabla f(\mathrm{w}_t)\rangle = \langle\mathrm{w}^* - \mathrm{w}_t, \nabla f(\mathrm{w}_t)\rangle + \langle\mathrm{w}_t - \tilde{\mathrm{w}}_t, \nabla f(\mathrm{w}_t)\rangle$$

The first term is bounded by $\mu$-strong convexity as,

$$f(\mathrm{w}^*) \geq f(\mathrm{w}_t) + \langle \nabla f(\mathrm{w}_t), \mathrm{w}^* - \mathrm{w}_t \rangle + \frac{\mu}{2} \|\mathrm{w}_t - \mathrm{w}^*\|^2$$

$$\iff \langle \nabla f(\mathrm{w}_t), \mathrm{w}^* - \mathrm{w}_t \rangle \leq f(\mathrm{w}^*) - f(\mathrm{w}_t) - \frac{\mu}{2} \|\mathrm{w}_t - \mathrm{w}^*\|^2$$

$$\leq -\epsilon_t + \frac{\mu}{2}\mu \|\tilde{\mathrm{w}}_t - \mathrm{w}_t\| - \frac{\mu}{4} \|\mathrm{w}^* - \tilde{\mathrm{w}}_t\|$$

where in the last step, we define $\epsilon_t := f(\mathrm{w}_t) - f(\mathrm{w}^*)$ and use $\|\mathrm{u} + \mathrm{v}\|^2 \leq 2(\|\mathrm{u}\|^2 + \|\mathrm{v}\|^2)$. The second term is bounded by using $2\langle \mathrm{u}, \mathrm{v} \rangle \leq a \|\mathrm{u}\|^2 + \frac{1}{a} \|\mathrm{v}\|^2$ as follows,

$$2 \langle \mathrm{w}_t - \tilde{\mathrm{w}}_t, \nabla f(\mathrm{w}_t) \rangle \leq 2L \|\mathrm{w}_t - \tilde{\mathrm{w}}_t\|^2 + \frac{1}{2L} \|\nabla f(\mathrm{w}_t)\|^2$$

Moreover, from Facr 2, we have $\|\nabla f(\mathrm{w}_t)\|^2 \leq 2L\epsilon_t$. Taking expectation and putting everything together, we get,

$$\mathbb{E}\left[\|\tilde{\mathrm{w}}_{t+1} - \mathrm{w}^*\|^2\right] \leq \left(1 - \frac{\mu\eta_t}{2}\right) \mathbb{E}\left[\|\tilde{\mathrm{w}}_t - \mathrm{w}^*\|^2\right] + \eta_t^2 \sigma^2$$

$$+ (\mu + 2L)\,\eta_t \mathbb{E}\left[\|\mathrm{w}_t - \tilde{\mathrm{w}}_t\|^2\right] + \left(2L\eta_t^2 - \eta_t\right)\epsilon_t$$

We now claim that the last term $2L\eta_t^2 - \eta_t \leq -\frac{\xi-2}{\xi}\eta_t$ or equivalently $2L\eta_t^2 - \left(1 - \frac{\xi-2}{\xi}\right)\eta_t \leq 0$. Note that this is a quadratic in $\eta_t$ which is satisfied between its roots $0$ and $\frac{1}{L\xi}$. So it suffices to show is that our step sizes are in this range. In particular, the second root (which is positive by choice of $\xi$) should be no less than step size. We have $\eta_t = \frac{1}{\mu(t+\xi)}$, $\eta_t \leq \frac{1}{\mu\xi} \ \forall\ t$, the second root $\frac{1}{L\xi} \geq \frac{1}{\mu\xi}$ because smoothness parameter $L \geq \mu$, the strong convexity parameter, or equivalently the condition number $\kappa := L/\mu \geq 1$. Combining the above with $\mathrm{a}_t = \mathrm{w}_t - \tilde{\mathrm{w}}_t$, we get,

$$\mathbb{E}\left[\|\tilde{\mathrm{w}}_{t+1} - \mathrm{w}^*\|^2\right] \leq \left(1 - \frac{\mu\eta_t}{2}\right) \mathbb{E}\left[\|\tilde{\mathrm{w}}_t - \mathrm{w}^*\|\right] + \eta_t^2 \sigma^2$$

$$+ (\mu + 2L)\,\eta_t \mathbb{E}\left[\|\mathrm{a}_t\|^2\right] - \left(1 - \frac{2}{\xi}\right)\eta_t\epsilon_t$$

$$\square$$

**Fact 1.** *Rakhlin et al. [2012] Let $f : \mathbb{R}^d \to$ be a $\mu$-strongly convex function, and $\mathrm{w}^*$ be its minima. Let $\mathrm{g}$ be an unbiased stochastic gradient at point $\mathrm{w}$ such that $\mathbb{E}\left[\|\mathrm{g}\|^2\right] \leq G^2$, then*

$$\mathbb{E}\left[\|\mathrm{w} - \mathrm{w}^*\|^2\right] \leq \frac{4G^2}{\mu^2}$$

**Fact 2.** *For $L$-smooth convex function $f$ with minima $\mathrm{w}^*$, then the following holds for all points $\mathrm{w}$,*

$$\|\nabla f(\mathrm{w}) - \nabla f(\mathrm{w}^*)\|^2 \leq 2L(f(\mathrm{w}) - f(\mathrm{w}^*))$$

**Fact 3.** *Stich et al. [2018] Let $\{b_t\}_{t\geq 0}, b_t \geq 0$ and $\{\epsilon_t\}_{t\geq 0}, \epsilon_t \geq 0$ be sequences such that,*

$$b_{t+1} \leq \left(1 - \frac{\mu\eta_t}{2}\right) b_t - \epsilon_t\eta_t + A\eta^2 + B\eta^3$$

*for constants $A, B > 0, \mu \geq 0, \xi > 1$. Then,*

$$\frac{1}{Q_T} \sum_{t=0}^{T-1} q_t\epsilon_t \leq \frac{\mu\xi^3 b_0}{8Q_T} + \frac{4T(T+2\xi)A}{\mu Q_T} + \frac{64TB}{\mu^2 Q_T}$$

*for $\eta_t = \frac{8}{\mu(\xi+t)}, q_t = (\xi+t)^2, Q_T = \sum_{t=0}^{T-1} q_t \geq \frac{T^3}{3}$*

**Fact 4.** *Stich et al. [2018] Let $\{h_t\}_{t>0}$ be a sequence satisfying $h_0 = 0$ and*

$$h_{t+1} \leq \min\left\{ (1 - \tau/2) h_t + \frac{2}{\tau_k}\eta_t^2 A, (t+1)\sum_{i=0}^{t} \eta_i^2 A \right\}$$

*for constant $A > 0$, then with $\eta_t = \frac{1}{t+\xi}$ with $\xi > 1 + \frac{1+\beta}{\tau_k(1+\rho)}$, with $\beta > 4$ and $\rho = \frac{4\beta}{(\beta-4)(\beta+1)^2}$, for $t \geq 0$ we get,*

$$h_t \leq \frac{4\beta}{(\beta - 4)} \cdot \frac{\eta_t^2 A}{\tau_k^2}$$

**Lemma 3.** *With probability at least $1 - \delta$*

$$\mathbb{E}\left[\|\mathbf{a}_t\|^2\right] \leq \frac{4\beta}{(\beta - 4)} \cdot \frac{\eta_t^2 G^2}{\tau_k^2}$$

*Proof of Lemma 3.* The proof repeats the steps in Stich et al. [2018] with minor modifications. In particular, the compression is provided by the recovery guarantees of Count Sketch, and we do a union bound over all its instances. We write the proof in full for the sake of completeness. Note that

$$\mathbf{a}_t = \mathbf{a}_{t-1} + \eta_{t-1}\mathbf{g}_{t-1} - \tilde{\mathbf{g}}_{t-1}$$

We first claim that $\mathbb{E}\left[\|\mathbf{a}_t\|^2\right] \leq t\eta_t^2 G^2$. Since $\mathbf{a}_0 = 0$, we have $\mathbf{a}_t = \sum_{i=1}^{t}(\mathbf{a}_i - \mathbf{a}_{i-1}) = \sum_{i=0}^{t-1}(\eta_i\mathbf{g}_i - \tilde{\mathbf{g}}_i)$. Using $(\sum_{i=1}^{n} a_i)^2 \leq (n+1)\sum_{i=1}^{n} a_i^2$ and taking expectation, we have

$$\mathbb{E}\left[\|\mathbf{a}_t\|^2\right] \leq t\sum_{i=0}^{t-1}\mathbb{E}\left[\|\eta_i\mathbf{g}_i - \tilde{\mathbf{g}}_i\|^2\right] \leq t\sum_{i=0}^{t-1}\eta_i^2 G^2$$

Also, from the guarantee of Count Sketch, we have that, with probability at least $1 - \delta/T$, the following holds give that our compression is a $\tau_k$ contraction.

Therefore

$$\|\mathbf{a}_{t+1}\|^2 \leq (1 - \tau_k)\|\mathbf{a}_t + \eta_t\mathbf{g}_t\|^2$$

Using inequality $(a + b)^2 \leq (1 + \gamma)a^2 + (1 + \gamma^{-1})b^2, \gamma > 0$ with $\gamma = \frac{\tau_k}{2}$, we get

$$\|\mathbf{a}_{t+1}\|^2 \leq \tau_k\left((1 + \gamma)\|\mathbf{a}_t\|^2 + \left(1 + \gamma^{-1}\right)\eta_t^2\|\mathbf{g}_t\|^2\right)$$

$$\leq \frac{(2 - \tau_k)}{2}\|\mathbf{a}_{t-1}\|^2 + \frac{2}{\tau_k}\eta_t^2\|\mathbf{g}_t\|^2$$

Taking expectation on the randomness of the stochastic gradient oracle, and using $\mathbb{E}\left[\|\mathbf{g}_t\|^2\right] \leq G^2$, we have,

$$\mathbb{E}\left[\|\mathbf{a}_{t+1}\|^2\right] \leq \frac{(2 - \tau_k)}{2}\mathbb{E}\left[\|\mathbf{a}_t\|^2\right] + \frac{2}{\tau_k}\eta_t^2 G^2$$

Note that for a fixed $t \leq T$ this recurrence holds with probability at least $1 - \delta/T$. Using a union bound, this holds for all $t \in [T]$ with probability at least $1 - \delta$. Conditioning on this and using Fact 4 completes the proof. □

# B Auxiliary results

We state the result of Stich et al. [2018] in full here.

**Fact 5** ([Stich et al., 2018]). *Let $f : \mathbb{R}^d \to \mathbb{R}$ be a L-smooth $\mu$-strongly convex function. Given $T > 0$ and $0 < k \le d$, sparsified SGD with step size $\eta_t = \frac{1}{t+\xi}$, with $\xi > 1 + \frac{d(1+\beta)}{k(1+\rho)}$, with $\beta > 4$ and $\rho = \frac{4\beta}{(\beta-4)(\beta+1)^2}$, after $T$ steps outputs $\hat{\mathrm{w}}_T$:*

$$\mathbb{E}\left[f(\hat{\mathrm{w}}_T)\right] - f(\mathrm{w}^*) \le \mathcal{O}\left(\frac{G^2}{\mu T} + \frac{d^2 G^2 L}{k^2 \mu^2 T^2} + \frac{d^3 G^3}{k^3 \mu T^3}\right).$$

We now state theorem which uses on the norm bound on stochastic gradients. It follows by directly plugging the fact the HEAVYMIX is a $k/d$-contraction in the result of Stich et al. [2018].

**Theorem 3.** *Let $f : \mathbb{R}^d \to \mathbb{R}$ be a L-smooth $\mu$-strongly convex function . Given $T > 0$ and $0 < k \le d, 0 < \delta < 1$, Algorithm 2 on one machine, with access to stochastic gradients such that $\mathbb{E}\left[\|\mathrm{g}\|^2\right] \le G^2$, with sketch size $\mathcal{O}\left(k \log(dT/\delta)\right)$ and step size $\eta_t = \frac{1}{t+\xi}$, with $\xi > 1 + \frac{d(1+\beta)}{k(1+\rho)}$, with $\beta > 4$ and $\rho = \frac{4\beta}{(\beta-4)(\beta+1)^2}$, after $T$ steps outputs $\hat{\mathrm{w}}_T$ such that with probability at least $1 - \delta$:*

$$\mathbb{E}\left[f(\hat{\mathrm{w}}_T)\right] - f(\mathrm{w}^*) \le \mathcal{O}\left(\frac{G^2}{\mu T} + \frac{d^2 G^2 L}{k^2 \mu^2 T^2} + \frac{d^3 G^3}{k^3 \mu T^3}\right).$$

**Theorem 4** ((non-convex, smooth)). *Let $\{\mathrm{w}_t\}_{t \ge 0}$ denote the iterates of Algorithm 2 one one machine, on an L-smooth function $f : \mathbb{R}^d \to \mathbb{R}$. Assume the stochastic gradients g satisfy $\mathbb{E}[\mathrm{g}] = \nabla f(\mathrm{w})$ and $\mathbb{E}[\|\mathrm{g}\|_2^2] \le G^2$, and use a sketch of size $\mathcal{O}(k \log(dT/\delta))$, for $0 \le \delta \le 1$. Then, setting $\eta = 1/\sqrt{T+1}$ with probability at least $1 - \delta$:*

$$\min_{t \in [T]} \|\nabla f(\mathrm{w}_t)\|^2 \le \frac{2 f_0}{\sqrt{(T+1)}} + \frac{L G^2}{2\sqrt{T+1}} + \frac{4 L^2 G^2 (1 - k/d)}{(k/d)^2 (T+1)},$$

*where $f_0 = f(\mathrm{w}_0) - f^\star$.*

**Theorem 5** ((convex, non-smooth)). *Let $\{\mathrm{w}_t\}_{t \ge 0}$ denote the iterates of Algorithm 2 one one machine, on a convex function $f : \mathbb{R}^d \to \mathbb{R}$. Define $\bar{\mathrm{w}}_\mathrm{t} = \frac{1}{T}\sum_{t=0}^{T} \mathrm{w}_t$. Assume the stochastic gradients g satisfy $\mathbb{E}[\mathrm{g}] = \nabla f(\mathrm{w})$ and $\mathbb{E}[\|\mathrm{g}\|_2^2] \le G^2$, and use a sketch of size $\mathcal{O}(k \log(dT/\delta))$, for $0 \le \delta \le 1$. Then, setting $\eta = 1/\sqrt{T+1}$, with probability at least $1 - \delta$:*

$$\mathbb{E}[f(\bar{\mathrm{w}}_\mathrm{t})] - f^\star \le \frac{\|\mathrm{w}_0 - \mathrm{w}^\star\|^2}{\sqrt{(T+1)}} + \left(1 + \frac{2\sqrt{1-k/d}}{k/d}\right)\frac{G^2}{\sqrt{T+1}}.$$

Our high probability bounds of Theorem 2 can be converted to bounds in expectation, stated below.

**Theorem 6.** *Let $f : \mathbb{R}^d \to \mathbb{R}$ be a L-smooth $\mu$-strongly convex function. Given $T > 0$ and $0 < k \le d, 0 < \delta < 1$, and a $\tau_k$-contraction, Algorithm 2 one one machine, with sketch size $\mathcal{O}\left(k \log(dT/\delta)\right)$ and step size $\eta_t = \frac{1}{t+\xi}$, with $\xi > 1 + \frac{1+\beta}{\tau_k(1+\rho)}$, with $\beta > 4$ and $\rho = \frac{4\beta}{(\beta-4)(\beta+1)^2}$ and $\delta = \mathcal{O}\left(\frac{k}{poly(d)}\right)$ after $T$ steps outputs $\hat{\mathrm{w}}_T$ such that*

$$\mathbb{E}_{\mathcal{A}}\mathbb{E}\left[f(\hat{\mathrm{w}}_T)\right] - f(\mathrm{w}^*) \le \mathcal{O}\left(\frac{\sigma^2}{\mu T} + \frac{G^2 L}{\tau_k^2 \mu^2 T^2} + \frac{G^3}{\tau_k^3 \mu T^3}\right)$$

*Proof.* Lemma 1 gives that with probability at least $1 - \delta$, HEAVYMIX is a $k/d$ contraction. We leverage the fact that the elements of countsketch matrix are bounded to convert it to bound in expectation. As in the proof of lemma 1, given $\mathrm{g} \in \mathbb{R}$, the HEAVYMIX algorithm extracts all $(1/k, \ell_2^2)$-heavy elements from a Count Sketch $S$ of g. Let $\hat{\mathrm{g}}$ be the values of all elements recovered from sketch. For a fixed $k$, we create two sets $H$ (heavy), and $NH$ (not-heavy). All coordinates of $\hat{\mathrm{g}}$ with values at least $\frac{1}{k}\hat{\ell}_2^2$ are put in $H$, and all others in $NH$, where $\hat{\ell}_2$ is the estimate of $\|\mathrm{g}\|_2$

(a) Low level intuition behind the update step of the Count Sketch.

(b) Property of mergeability lets the parameter server approximate the heavy coordinates of the aggregate vector

from the Count Sketch. For a $\tau_k$ contraction with probability at least $1 - \delta$, we get the following expectation bound.

$$\mathbb{E}_{\mathcal{A}}\mathbb{E}\left[\|g - \bar{g}\|^2\right] \leq (1 - \delta)(1 - \tau_k)\|g\|^2 + \delta\mathcal{O}(\text{poly}(d))\|g\|^2$$

$$\leq (1 - \frac{\tau_k}{2})\|g\|^2$$

where the last time follows because we choose $\delta = \frac{\tau_k}{2\mathcal{O}(\text{poly}(d))}$. Since HEAVYMIX is a $k/d$ contraction, we get the expectation bound of $k/2d$ with $\delta = \frac{k}{2d\mathcal{O}(\text{poly}(d))}$

$\square$

## C  Sketching

Sketching gained its fame in the streaming model [Muthukrishnan et al., 2005]. A seminal paper by Alon et al. [1999] formalizes the model and delivers a series of important results, among which is the $\ell_2$-norm sketch (later referred to as the AMS sketch). Given a stream of updates $(a_i, w_i)$ to the $d$ dimensional vector g (i.e. the $i$-th update is $g_{a_i} += w_i$), the AMS sketch initializes a vector of random signs: $s = (s_j)_{j=1}^d, s_j = \pm 1$. On each update $(a_i, w_i)$, it maintains the running sum $S += s_{a_i}w_i$, and at the end it reports $S^2$. Note that, if $s_j$ are at least 2-wise independent, then $E(S^2) = E(\sum_i g_i s_i)^2 = \sum_i g_i^2 = \|g\|_2^2$. Similarly, the authors show that 4-wise independence is enough to bound the variance by $4\|g\|_2^2$. Averaging over independent repetitions running in parallel provides control over the variance, while the median filter (i.e. the majority vote) controls the probability of failure. Formally, the result can be summarized as follows: AMS sketch, with a large constant probability, finds $\hat{\ell}_2 = \|g\|_2 \pm \varepsilon\|g\|_2$ using only $\mathcal{O}\left(\frac{1}{\varepsilon^2}\right)$ space. Note that one does not need to explicitly store the entire vector $s$, as its values can be generated on thy fly using 4-wise independent hashing.

**Definition 3.** *Let* g $\in \mathbb{R}^d$. *The $i$-th coordinate $g_i$ of g is an $(\alpha_1, \ell_2)$-heavy hitter if $|g_i| \geq \alpha_1\|g\|_2$. $g_i$ is an $(\alpha_2, \ell_2^2)$-heavy hitter if $g_i^2 \geq \alpha_2\|g\|_2^2$.*

The AMS sketch was later extended by Charikar et al. [2002] to detect heavy coordinates of the vector (see Definition 3). The resulting Count Sketch algorithm hashes the coordinates into $b$ buckets, and sketches the $\ell_2$ norm of each bucket. Assuming the histogram of the vector values is skewed, only a small number of buckets will have relatively large $\ell_2$ norm. Intuitively, those buckets contain the heavy coordinates and therefore all coordinates hashed to other buckets can be discarded. Repeat the same routine independently and in parallel $\mathcal{O}(\log_b d)$ times, and all items except the heavy ones will be excluded. Details on how to combine proposed hashing and $\ell_2$ sketching efficiently are presented in Figure 5a and Algorithm 4.

Count Sketch finds all $(\alpha, \ell_2)$-heavy coordinates and approximates their values with error $\pm\varepsilon\|g\|_2$. It does so with a memory footprint of $\mathcal{O}\left(\frac{1}{\varepsilon^2\alpha^2}\log d\right)$. We are more interested in finding $(\alpha, \ell_2^2)$-heavy hitters, which, by an adjustment to Theorem 7, the Count Sketch can approximately find with a space complexity of $\mathcal{O}\left(\frac{1}{\alpha}\log d\right)$, or $\mathcal{O}(k \log d)$ if we choose $\alpha = \mathcal{O}\left(\frac{1}{k}\right)$.

Both the Count Sketch and the Count-Min Sketch, which is a similar algorithm presented by Cormode and Muthukrishnan [2005] that achieves a $\pm\varepsilon\ell_1$ guarantee, gained popularity in distributed systems primarily due to the mergeability property formally defined by Agarwal et al. [2013]: given a sketch $S(f)$ computed on the input vector $f$ and a sketch $S(g)$ computed on input $g$, there exists a function $F$, s.t. $F(S(f), S(g))$ has the same approximation guarantees and the same memory footprint as $S(f+g)$. Note that sketching the entire vector can be rewritten as a linear operation $S(f) = Af$, and therefore $S(f+g) = S(f) + S(g)$. We take advantage of this crucial property in SKETCHED-SGD, since, on the parameter server, the sum of the workers' sketches is identical to the sketch that would have been produced with only a single worker operating on the entire batch.

Besides having sublinear memory footprint and mergeability, the Count Sketch is simple to implement and straight-forward to parallellize, facilitating GPU acceleration [Ivkin et al., 2018].

Charikar et al. [2002] define the following approximation scheme for finding the list $T$ of the top-$k$ coordinates: $\forall i \in [d] : i \in T \Rightarrow g_i \geq (1-\varepsilon)\theta$ and $g_i \geq (1+\varepsilon)\theta \Rightarrow i \in T$, where $\theta$ is chosen to be the $k$-th largest value of $f$.

**Theorem 7** (Charikar et al., 2002). *Count Sketch algorithm finds approximate top-$k$ coordinates with probability at least $1-\delta$, in space $O\left(\log\frac{d}{\delta}\left(k + \frac{\|\mathrm{g}^{tail}\|_2^2}{(\varepsilon\theta)^2}\right)\right)$, where $\|\mathrm{g}^{tail}\|_2^2 = \sum_{i \notin top\ k} \mathrm{g}_i^2$ and $\theta$ is the $k$-th largest coordinate.*

Note that, if $\theta = \alpha\|\mathrm{g}\|_2$, Count Sketch finds all $(\alpha, \ell_2)$-heavy coordinates and approximates their values with error $\pm\varepsilon\|\mathrm{g}\|_2$. It does so with a memory footprint of $\mathcal{O}\left(\frac{1}{\varepsilon^2\alpha^2}\log d\right)$.

---

**Algorithm 4** Count Sketch [Charikar et al., 2002]

---

```
 1: function init(r, c):
 2:     init sign hashes {s_j}^r_{j=1} and bucket hashes {h_j}^r_{j=1}
 3:     init r × c table of counters S
 4: function update(i, f_i):
 5:     for j in 1 . . . r:
 6:         S[j, h_j(i)] += s_j(i)f_i
 7: function estimate(i):
 8:     init length r array estimates
 9:     for j in 1, . . . , r:
10:         estimates[r] = s_j(i)S[j, h_j(i)]
11:     return median(estimates)
```

---

# D    Model Training Details

We train three models on two datasets. For the first two models, we use code from the OpenNMT project Klein et al. [2017], modified only to add functionality for SKETCHED-SGD. The command to reproduce the baseline transformer results is

```
python  train.py -data $DATA_DIR -save_model baseline -world_size 1
        -gpu_ranks 0 -layers 6 -rnn_size 512 -word_vec_size 512
        -batch_type tokens -batch_size 1024 -train_steps 60000
        -max_generator_batches 0 -normalization tokens -dropout 0.1
        -accum_count 4 -max_grad_norm 0 -optim sgd -encoder_type transformer
        -decoder_type transformer -position_encoding -param_init 0
        -warmup_steps 16000  -learning_rate 1000 -param_init_glorot
        -momentum 0.9 -decay_method noam -label_smoothing 0.1
        -report_every 100 -valid_steps 100
```

The command to reproduce the baseline LSTM results is

```
python  train.py -data $DATA_DIR -save_model sketched -world_size 1
        -gpu_ranks 0 -layers 6 -rnn_size 512 -word_vec_size 512
        -batch_type tokens -batch_size 1024 -train_steps 60000
        -max_generator_batches 0 -normalization tokens -dropout 0.1
        -accum_count 4 -max_grad_norm 0 -optim sgd -encoder_type rnn
```

(a) log-log plot of training and test error against number of iterations of the average iterate for SVM trained on one class as positive and the rest as negative (1-v-all). For simplicity, we only show the plot for one class.

(b) log-log plot of training and test error of the number of iterations for regularized logistic regression. The regularization parameter was fixed as 0.01.

```
-decoder_type rnn -rnn_type LSTM -position_encoding -param_init 0
-warmup_steps 16000 -learning_rate 8000 -param_init_glorot
-momentum 0.9 -decay_method noam -label_smoothing 0.1
-report_every 100 -valid_steps 100
```

We run both models on the WMT 2014 English to German translation task, preprocessed with a standard tokenizer and then shuffled.

The last model is a residual network trained on CIFAR-10. We use the model from the winning entry of the DAWNBench competition in the category of fastest training time on CIFAR-10 Coleman et al. [2017]. We train this model with a batch size of 512, a learning rate varying linearly at each iteration from 0 (beginning of training) to 0.4 (epoch 5) back to 0 (end of training). We augment the training data by padding images with a 4-pixel black border, then cropping randomly back to 32x32, making 8x8 random black cutouts, and randomly flipping images horizontally. We use a cross-entropy loss with L2 regularization of magnitude 0.0005.

Each run is carried out on a single GPU – either a Titan X, Titan Xp, Titan V, Tesla P100, or Tesla V100.

# E   Additional experiments

## E.1   MNIST

We train vanilla and sketched counterparts of two simple learning models: Support vector machines(SVM) and $\ell_2$ regularized logistic regression on MNIST dataset. These are examples of optimizing non-smooth convex function and strongly convex smooth function respectively. We also compare against the theoretical rates obtained in Theorems 5 and 1. The sketch size used in these experiments is size 280 (40 columns and 7 rows), and the parameters $k$ and $P$ are set as, $k = 10, P = 10$, giving a compression of around 4; the number of workers is 4. Figure 5a and 5b shows the plots of training and test errors of these two models. In both the plots, we see that the train and test errors decreases with $T$ in the same rates for vanilla and sketched models. However, these are conservative compared to the theoretical rate suggested.