[Reviews · NeurIPS 2019]

Reviewer 1



Overall, the paper develops a new and practically useful algorithm for the distributed SGD problem with theoretical convergence guarantees and empirical evidences on its performance. The paper can be improved in terms of clarity by expanding on the main theoretical advantages of the algorithm over the Stich et al. paper and addressing some concerns on the Count Sketch detailed below.

Reviewer 2



The authors propose sketching gradients to solve the communication overhead of large-scale distributed SGD. In particular, they show a reduction from O(d) to O(log d) where d is number of model parameters. Count-sketch is used to estimate the top-k elements of the gradient by first identifying the heavy-hitters and then getting their values from the worker nodes. Extensive experimental results are shown on a wide-variety of tasks showing up to 40x improvement in communication cost. Comments: Overall, the paper is well-written with clear remarks to show how it differs from related work such as Stich et al which it heavily relies on. The results are important because SGD is a workhorse for a wide-range of deep learning models and speeding up the communication can help train better models/architectures in a shorter time.

Reviewer 3



|------------------------------> The authors have addressed my questions in the rebuttal, and so I am increasing my score from a 6 to a 7. |------------------------------| This work, to my knowledge, appears to be original. The quality is adequate; the authors show familiarity with, and build on ideas from, the relevant literature. The experimental setup (image classification and NMT) is also relevant. The work is very clear and well written. The proposed method could provide a significant reduction in training time for practitioners and researchers, but, in my opinion, needs some additional empirical validation. Criticisms: 1. The bounded second moment and variance assumptions, together, are quite strong. There has been a lot of work recently advocating against the use of these combined assumptions. 2. Communication complexity per iteration is "k log(dT/delta)W"; but k log(dT/delta) > d for sufficiently large T. So does this mean that one can only use gradient sketches for the first few iterations of training? 3. In related work, it is stated that other gradient compression methods that "achieve high compression rates when the workers transmit gradients to the parameter server," still incur O(w) communication overhead when the model parameters are communicated back, but this seems like a technicality of the exposition and not of the method. Couldn't one just send back the compressed gradient in this case and have the workers apply updates locally, or, alternatively, only return the updated coordinates? 4. In line 8, in Algorithm 2, you only send back the k elements that were modified right? Worth stating in the algorithm description itself. 5. You left a "TODO" in table 1. 6. To competitively train a transformer on WMT En-De, one typically uses adaptive step-size methods like Adam, not vanilla SGD. How does the gradient Sketching approach in parameter-server setting extend to cases like Adam, where the momentum buffers are non-linear in the stochastic gradients? It seemed like the linearity property of the Sketches was essential, and so it's not clear to me that this method would work with methods like Adam. 7. Does "vanilla SGD" in the experiments also contain some form of momentum? 8. Can you explain why momentum factor masking "interferes with momentum" such that increasing k (reducing compression error) increases the training error? 9. Given that your method can achieve significant communication reduction at the expense of a reduction in model accuracy, can you train for more iterations and obtain better models while still maintaining less communication overall? 10. How does this method compare, empirically, with other compression schemes.

[Author Response · NeurIPS 2019]

**Reviewer 1. 1** *."more background review"* - Thanks for the references. We will add discussion of these papers. **2.*"return communication grows with W.."*** - In $k$-sparsification methods, the weight update is the sum of $W$ $k$-sparsified gradients and can have $Wk$ non-zero entries. In practice, the weight update is dense for large $W$, as shown in Figure 4. There is no way to losslessly compress this back-communication to under $O(Wk)$ bytes, and we are not aware of any lossy compression that achieves good performance in practice. **3.** *"In practice how should the size of the Count Sketch table be set?"* - Theory suggests the sketch size should be $O(k \log d)$, with $c = O(k)$ columns and $r = O(\log_c d)$ rows. The exact values must be found experimentally. *"I expect if the model parameters themselves are sparse then the gradients are more likely to be sparse"* - In general, sparsity of model parameters doesn't imply sparsity of gradients. For example, consider squared loss $(y - w^\top x)^2$ with a sparse w, the gradient is $-2(y - w^\top x)x$, which might not be sparse. We make no assumptions about the sparsity of the gradients, and for the models and loss functions we use (and that are commonly used), the gradients are dense.

**Reviewer 2.(A)** *"does it reduce training time.."* - We have not yet benchmarked overall running time on realistic network topologies, though we are actively working in this direction. Sketching does impose some computational overhead, and with our current implementation we expect a speedup in overall run time when communication cost is about as high (or higher) than the computation cost. This is realistic for a large network ( 200M parameters) even over a fast (10Gbps) network. In the regime of many workers (e.g. federated learning), where our algorithm performs best theoretically, network can be as slow as a few Mbps. **(B)** *using asynchronous updates* - We have not experimented with asynchronous updates, but we expect that the benefits asynchrony brings to regular SGD would also apply to Sketched SGD. We believe our theoretical results could be extended to the asynchronous case, but we leave this to future work.

**Reviewer 3.** 1. *"Bounded second moment and variance assumption"* - A bounded second moment implies bounded variance (via variational definition of variance) , so $\sigma \leq B$. However note that when you average $W$ stochastic gradients, the variance reduces by factor $W$, but the second moment bound is still the same. We use different parameters for variance & second moment bounds so that it is easy to argue how the quantities in the upper bound change when we average stochastic gradients. See remark 3 in the paper.

2.*"$k \log(dT/\delta) > d$ for sufficiently large T..."* - $T > \Omega(e^d)$ is not realistic for even our smaller model ($d \approx 10^7$).

3. *"Can't one send the compressed gradients back..?"* The $O(W)$ per-worker communication cost is not a technicality: if each worker sends $k$ non-zero gradient elements, the weight update (which is the sum of these sparse gradients) will have up to $kW$ non-zero elements. In practice, the weight update quickly does become dense, as shown in Figure 4. In principle, one could lossily compress the back communication, but we are not aware of any methods that do so while maintaining high performance.

4. *".. you only send back the k elements that were modified right?"* - Correct, $\tilde{g}_t$ only has k non-zero elements, so we only need to transmit $O(k)$ bytes. We will add a remark to make this clear. 5. *"TODO"* - Thanks!

6. *"extends to methods like Adam?"* - This is a good point. The Adam update uses the $\ell_2$ norm of the sum (or linear combination) of gradients in the update, and count sketch can be used to approximate the $\ell_2$ norm. Therefore we can, in principle, mimic the Adam update. We do not know if this approximation is good enough to preserve Adam's theoretical properties, or if it would work well in practice.

7. *Regarding momentum?* - Sorry for the poor terminology – all experiments use momentum.

8. *"..why momentum factor masking 'interferes with momentum'"* - in line 11 of Algorithm 3, we zero out the coordinates of each $u_i$ that were updated (this is momentum factor masking). In the limit of $k \rightarrow d$, each coordinate is updated every iteration, so there is effectively no momentum.

9.*"communication reduction at the expense of model accuracy"* - To be clear, we get no drop in performance with 40x compression for the translation task. For CIFAR-10, we can make up the accuracy drop by training longer. For example, training with 17x compression for the usual number of iterations gives 92.5% validation accuracy. Training with 50% more iterations (reducing to 11x overall compression) restores accuracy to 94%. We expect similar results to hold for the translation task at compression rates higher than 40x.

10. *"compare empirically against others"* - We outperform many popular methods. For example, the "Local Top-$k$" method in Figure 4 is very similar to deep gradient compression (Lin+2017), except we do not clip gradients, and we warm up the learning rate instead of the sparsity. Local Top-$k$ does about as well as 9x-compression Sketched SGD, but it achieves only $\sim 2x$ compression for large enough $W$. We also implemented and compared to signSGD (arXiv:1810.05291), which gets 92.5% validation and 32x compression. When we quantize all communication in Sketched SGD to 16 bits, we achieve 92.5% accuracy with 34x compression ($r = 3$, $c = 100,000$, $P = 8$). For the translation task, we already achieve 40x compression with no loss in accuracy, which is better than signSGD's 32x, and which we expect to double to 80x without loss of accuracy by adding 16-bit quantization. TernGrad (arxiv:1705.07878) is another popular method, but achieves hardly any compression of the back-communication from parameter server to workers for the large models we consider with $d \approx 2^{26}$ – see TernGrad pg. 3.

[Meta-Review · NeurIPS 2019]

There is a consensus among reviewers that this paper is well above the acceptance threshold. The importance of the problem and simplicity of methods would be an enjoyable read for the audience. Authors are encouraged to take reviews into account before preparing the final version.